# Exploring the Association Between Adolescents’ Health Literacy and Health Behavior by Using the Short Health Literacy (HLS_19_-Q12) Questionnaire

**DOI:** 10.3390/healthcare12242585

**Published:** 2024-12-22

**Authors:** Saulius Sukys, Gerda Kuzmarskiene, Kristina Motiejunaite

**Affiliations:** Department of Physical and Social Education, Lithuanian Sports University, Sporto g. 6, LT-44221 Kaunas, Lithuania; gerda.kuzmarskiene@stud.lsu.lt (G.K.); kristina.motiejunaite@lsu.lt (K.M.)

**Keywords:** adolescents, health literacy, HLS_19_-Q12, health behavior

## Abstract

Background: Health literacy (HL) is an important element of public health that is linked to health behavior in children and adolescents. This study aimed to investigate the structural validity and reliability of the HLS_19_-Q12 in the adolescent population, to measure the general HL of adolescents, and to assess the relationship between adolescents’ HL and health behavior. Methods: This cross-sectional study involved 825 students aged 15–19 years attending Lithuanian gymnasiums. The participants completed an online survey that collected information on key sociodemographic, HL, and health behavior indicators. Factor analysis, Cronbach’s alpha, and McDonald’s omega were used to validate the Lithuanian version of the HLS_19_-Q12. Health behavior indicators, including physical activity, smoking, alcohol consumption, self-rated health, and body mass index (BMI), were measured. We used regression analyses to assess the associations. Results: The HLS_19_-Q12 showed an acceptable reliability index (Cronbach’s α = 0.885, McDonald’s omega ω = 0.877) and adequate structural validity (comparative fit index  =  0.89, Tucker and Lewis’s index of fit  =  0.88, root mean square error of approximation  =  0.06). We found that 67.1% of the adolescents had excellent or sufficient HL, 27.7% had problematic HL, and 5.2% had inadequate HL. Compared with girls, boys had greater HL. HL was positively related to physical activity and self-rated health and negatively related to alcohol use and smoking. Conclusions: HLS_19_-Q12 is a reliable and valid measure of HL among adolescents in upper grades; higher levels of HL have been found to result in better health behaviors.

## 1. Introduction

Health literacy (HL) has been a growing issue since the term was first used in 1970 [1]. There are many definitions of HL. Malloy-Weir, with colleagues [2], counted as many as 250 of them, and there was considerable variation between them. This is because the concept of HL has evolved over time to meet the constant challenges of society. Currently, the definition of HL is broad and still expanding and includes information-seeking, decision-making, problem-solving, critical thinking, and communication [3], as well as a variety of social, personal, and cognitive skills that are essential for preserving, protecting, and enhancing personal and public health [4]. The HL can significantly influence a person’s health [5]. Studies have shown that insufficient HL can lead to lower adherence to medication regimens [6], lower knowledge of diseases and their management [7], worse treatment outcomes [8], and poorer quality of life [7,9]. Compared with those with adequate HL, those with insufficient HL are less likely to participate in preventive programs [9], have a higher risk for hospitalization [10], and have an increased risk of mortality [11]. This leads to the inefficient use of health services, increased healthcare costs, and health inequalities [6,12].

HL in children and adolescents is often described as an individual characteristic that shows how children and adolescents receive, perceive, evaluate, and communicate information about health and how it is used to make health-related decisions and adjust behavior [13]. Individual cognitive skills such as reading, writing, critical thinking, or information processing skills are the most emphasized [14], but factors such as self-reflection, self-efficacy, motivation, communication skills, or specific technical skills are often also included [13,15]. Paakkari and Paakkari [16] define HL in children and adolescents as a result of school health education that assesses students’ abilities in a given situation. Other authors emphasize how health-related information is used and applied in different life situations and social settings [1]. Studies have shown that inadequate HL among adolescents is associated with lower engagement in health-promoting behaviors [17], lower self-esteem [18], and poorer quality of life [14,19]. Adolescents with high levels of HL are more likely to take responsibility for their health [13,20], and they are more effective in accessing health-related information and using this knowledge to reinforce positive health behaviors [13]. High HL level is associated with a lower rate of being overweight [21] or underweight [22], more leisure-time physical activity [23,24], enhanced nutritional and oral hygiene practices [21], lower tobacco and alcohol consumption [17,24,25,26], better sleeping habits [26], better stress management [27], and higher academic achievement [28]. Adolescents with high levels of HL also perceive their health better than those with lower HL [29,30,31].

Studies in the last decade have revealed that a more in-depth analysis of HL is needed because of its value in improving adolescent health [17,32]. This requires valid and practical tools for measuring HL that allow objective and continuous assessment of adolescents’ HL and progress monitoring. HL assessment tools are validated for use in the adolescent population, but most tools measure only selective domains. Several systematic reviews have also shown this. Fleary and colleagues [17] identified seven HL measurement instruments and reported that these tools usually assess functional and media HL when measuring the relationship between adolescent HL and health behavior. Guo and colleagues [33] identified 29 HL measurement instruments for children and adolescents aged 6–24 years and emphasized that researchers have focused mainly on the functional domain. A systematic review by Okan and colleagues [34] identified 15 generic HL measurement instruments for children and adolescents and reported that measuring HL among children and adolescents is particularly difficult for several reasons. First, there is currently no universally recognized model or definition of HL, and it is highlighted as a significant weakness affecting the development of comparable methods to measure the concept accurately. Second, children’s cognitive and social development varies dramatically between age groups; therefore, it is essential to tailor concepts and their operationalization on the basis of age and developmental stage, considering the capabilities of children at various stages of childhood and adolescence. Researchers highlighted the need for further research to improve HL measures for children and adolescents [17,33,34].

In Lithuania, there is a lack of HL research; only a few HL surveys have been conducted in the adult population [35,36,37,38,39,40]. The HL of children and adolescents was measured using a brief Health Literacy for School-Aged Children (HLSAC) instrument with participants aged 13–16 years [41]. It should be noted that the HLSAC scale was created and assessed as part of the WHO Child and Adolescent Health Study, known as the HBSC (Health Behavior in School-Aged Children) [31]. Thus, this questionnaire is designed to measure the HL of younger adolescents. Although this questionnaire has been used with older adolescents (16–19 years) in other countries [42], data on its validity in this age group are lacking. No studies have measured HL in adolescents over 16 years of age in Lithuania. Although the HLS-EU-Q47 instrument, developed and validated by the HLS-EU consortium [43] to assess HL in individuals aged 15 and older, has been translated into Lithuanian [38] and could be used to evaluate the HL of adolescents aged 15 and above, its application among this age group remains uncertain. Several studies exploring the reliability of the HLS-EU-Q47 for measuring HL in adolescents over 15 years of age have questioned the validity of the instrument for this age group and have highlighted the need for age-appropriate HL assessment tools that are better aligned with adolescents’ developmental phase, interests, and experiences in the management of health-related information [44,45]. Thus, there is a need for a time-efficient and age- and ability-appropriate HL measurement instrument that can be used in adolescents aged 15 years and above.

On the basis of the HLS-EU instruments, the WHO Action Network on Measuring Population and Organizational Health Literacy (M-POHL) developed a 47-item instrument (HLS_19_-Q47) and adapted short forms—the HLS_19_-Q12 and the HLS_19_-Q16—to measure the general WHO European Region population HL [46]. Specific items in the questionnaires were modified and clarified by rephrasing or substituting those that were too complex or difficult to understand, thereby enhancing their comprehensibility for the general population [44,47]. To our knowledge, these HLS_19_ instruments have been used to measure and investigate HL and its domains in the adult population. Although the HLS_19_-YP12 questionnaire has been adapted for young people (16–25 years) [48], we believe that the HLS_19_-Q12 could also be appropriate for measuring HL in a young population because of its increased clarity and acceptability due to its brevity. Consequently, the research questions of the study presented in this article—to measure the HL of adolescents aged 15 years and above, and to determine how HL is related to health behavior—were investigated by using the HLS_19_-Q47 adapted short form, the HLS_19_-Q12, which is more transparent and more acceptable to adolescents due to its brevity. To answer these questions, we aimed (a) to examine the structural validity and reliability of the HLS_19_-Q12, (b) to measure the general HL of adolescents, and (c) to assess the relationship between adolescents’ HL and health behavior.

## 2. Materials and Methods

### 2.1. Study Design and Setting

This school-based, cross-sectional study was conducted in January and March 2024. The participants included in the survey were 9th-, 10th-, 11th- and 12th-grade students attending gymnasiums in Lithuania. A clustered hierarchical sampling design was used where the initial sampling unit was the school class.

### 2.2. Study Procedure

Ethical approval was obtained from the Lithuanian Sports University Social Research Ethics Committee (Protocol No. SMTEK-144, 30 November 2022). Permission to conduct the study was secured from the principals of the schools. With the help of class teachers and public health specialists at each school, information about the research and study aims was disseminated to parents or caregivers, who were asked for their consent for their students to participate. The participants were given the option to either agree or decline to participate in the survey by selecting the appropriate response on an online form: “I agree to participate” or “I disagree to participate”. Only those participants who agreed were provided with the study materials. Additionally, participants had the option to exit the online form at any time without submitting their responses. Data were gathered through an anonymous cross-sectional online survey form on the Google Forms platform. The students completed the questionnaire during class at school.

### 2.3. Study Participants

In total, 825 students from eight gymnasium schools in the Kaunas district participated in the study; 809 students completed questionnaires without any noticeable logical errors or missing items, yielding a response rate of 98.06%. The participants were selected via a clustered hierarchical sampling design, where the initial sampling unit was the school class. Among the 809 students (mean age: 16.40 ± 1.11 years), 57.8% (*n* = 468) were girls, whereas the remaining 42.2% (*n* = 341) were boys. By analyzing the data, girls were coded as 1, and boys were coded as 2.

### 2.4. Study Measures

#### 2.4.1. Sociodemographic Information

The sociodemographic indicators in this study included questions on gender, age, school grade, and the socioeconomic status of the family. The family’s socioeconomic status was measured using The Family Wealth Scale [49], which was used in previous studies with adolescents in Lithuania [25]. This scale includes six items that are associated with occupancy of bedrooms, dishwasher ownership, ownership of a car, number of computers, number of bathrooms, and holidays abroad. The respondents were divided into three family affluence groups: low, medium, and high.

#### 2.4.2. Health Literacy

In this study, we used the short-form version of the Health Literacy Survey (HLS_19_-Q12) to measure general HL from the Health Literacy Population Survey Project 2019–2021 [46]. After permission was obtained to use this questionnaire from the HLS_19_ Consortium, HLS_19_-Q12 was translated into Lithuanian according to the HLS_19_ study protocol. It should be noted that this questionnaire has already been adapted and validated in Lithuania with adults. More precisely, a study with a country’s physical education teachers confirmed the satisfactory structural validity (CFI = 0.924, TLI = 0.917, RMSEA = 0.081) of this measure [39]. Nevertheless, we conducted a pilot study (*n* = 45) to test the clarity of the translated version for adolescents. More specifically, this was performed to test the face validity of the HLS_19_-Q12. The potential participants were asked to rate how understandable and clear each statement was to them.

The general HL questionnaire (HLS_19_-Q12) scale consists of 12 statements and is scored on a 4-point Likert scale ranging from 1 to 4 (very easy to very difficult). The HLS_19_-Q12 showed adequate internal consistency, with an average Cronbach’s alpha of 0.78 across 17 countries [46].

By analyzing the data, the overall score of the HLS_19_-Q12 can be calculated as the percentage (ranging from 0–100) of items with valid responses that were answered with “very easy” or “easy” (i.e., the items were implicitly dichotomized) [46]. Higher score values indicate a higher level of general HL. The levels of the HL can also be calculated following the recommended procedures [46]. Four levels of HL were identified: excellent, sufficient, problematic, and inadequate.

#### 2.4.3. Health Behavior Indicators

Health behavior indicators, including physical activity, smoking, alcohol consumption, self-rated health, and body mass index (BMI), were measured. The participant’s physical activity was measured by asking, “How many days in the last 7 days did you exercise or engage in other physical activity that lasted at least 60 min per day?”. The answer options ranged from “none = 0” to “7 days = 7” [50]. For the smoking measure, respondents were requested to answer if they had ever smoked and, if so, for how many days they had smoked in their lifetime and in the last 30 days [50]. Similarly, they were requested to answer the following questions about alcohol consumption: “Have you ever drunk alcohol?”, “If so, how many days has it been like this in your lifetime and the last 30 days?” [50]. The response options for these questions were “Never = 1”, “1–2 days = 2”, “3–5 days = 3”, “6–9 days = 4”, “10–19 days = 5”, “20–29 days = 6”, and “30 and more days = 7”. Furthermore, to measure self-rated health, the students were asked the following question: Would you say your health is… with response options of poor (1), fair (2), good (3), and excellent (4) [51]. This single item was also used to study Lithuanian schoolchildren [50].

### 2.5. Data Analysis

All data analyses were performed using IBM SPSS statistics 29.0 (IBM, Armonk, NY, USA) and JASP 0.19.1.0 (University of Amsterdam, Amsterdam, The Netherlands). Prior to the main analysis, the structural validity of the HLS_19_-Q12 was examined with exploratory factor analyses (EFA) and confirmatory factor analyses (CFA). EFA and CFA with the same participants are not recommended [52], so the data were randomly divided into two subsamples. A factor analysis score of no less than 300 is recommended as a good number of participants [53]. In our study, 405 participants for the EFA and 404 participants for the CFA were sufficient.

To determine whether the data were suitable for EFA, the Kaiser–Meyer–Olkin (KMO) test and Bartlett’s test of sphericity were conducted. Data are considered suitable for factorability if the KMO is greater than 0.60 and Bartlett’s test is significant [54]. All variables’ loadings should be no less than 0.32 [53]. By conducting single-factor CFA, the following fit indices were set: the root mean square error of approximation (RMSEA), the comparative fit index (CTI), and the Tucker–Lewis index (TLI). The following target values are assumed to indicate a good data model fit [55]: a value of 0.90 or higher for CFI and TLI, as well as an RMSEA of 0.08 or lower [56]. The reliability of the HLS_19_-Q12 was assessed by calculating Cronbach’s alpha and McDonald’s omega (ω).

For descriptive statistics, the mean, standard deviation, sample size/proportion, and correlation were calculated. Asymmetry and Kurtosis coefficients were evaluated to check the normal distribution of the variables. The indicators of these coefficients do not exceed ±2 [57]. Therefore, parametric statistical analysis methods were used to compare the groups. Student’s t-test was used to compare two independent samples, and one-way analysis of variance (ANOVA) was used to compare more than two independent samples. Dependence between categorical variables was assessed via the chi-square (χ^2^) test. Differences were considered statistically significant when *p* < 0.05. The effect size for testing differences was measured via Cohen’s d and Eta squared [58].

A multinomial regression analysis was also performed (controlling for gender, school grade, and family affluence level indicators) to assess the associations of general HL with health behavior indicators (physical activity, lifetime and past 30 days of smoking, alcohol consumption, and self-rated health). The effect size of the predictors was measured via a standardized beta coefficient (β) [58].

## 3. Results

### 3.1. Sociodemographic Characteristics of Respondents

Among the 809 children included in this study, more than half were girls (57.8%). The mean age was 16.40 years (SD = 1.11). The largest group in the study was children from the 9th grade, and the smallest was from the 12th grade. Concerning socioeconomic status, half of the children were in the medium family affluence group (46.0%) (Table 1).

### 3.2. Descriptive Statistics for Predictive Variables

The descriptive statistics for the predictors revealed that boys were more physically active than girls (*t*(806) = −8.59, *p* < 0.001, *d* = 0.62) (Table 2). Analyses also showed that the self-rated health mean score for boys was significantly higher than that for girls (*t*(806) = −5.55, *p* < 0.001, *d* = 0.40). In addition, boys scored higher on the evaluation of personal health than girls did in all grades.

Regarding grade, statistically significant differences were found by comparing physical activity (*F*(3, 806) = 2.91, *p* = 0.034, η^2^ = 0.01). Tukey’s test revealed that 10th-grade children scored significantly higher than did 12th-grade children (*p* < 0.05). The results did not reveal significant differences compared to self-health evaluation by grade (*F* = 1.11, *p* = 0.344). Significant differences were found comparing smoking per lifetime (*F*(3, 806) = 16.78, *p* < 0.001, η^2^ = 0.06) and smoking during the past 30 days (*F*(3, 806) = 13.15, *p* < 0.001, η^2^ = 0.05) as well as alcohol use per lifetime (*F*(3, 806) = 26.20, *p* < 0.001, η^2^ = 0.09) and during the last 30 days (*F*(3, 806) = 9.93, *p* < 0.001, η^2^ = 0.04) by grade. In all instances, Tukey’s test revealed that 9th-, 10th-, and 11th-grade children differ from 12th-grade children (*p* < 0.05). Additionally, in all instances, except for alcohol use in the last 30 days, 9th-grade children statistically differed from 11th-grade children (*p* < 0.05), and 10th- and 11th-grade children statistically differed from 12th-grade children (*p* < 0.05).

Health behavior indicators were compared by family affluence (Table 2). The results revealed that there was a statistically significant difference between groups comparing physical activity (*F*(2, 807) = 6.72, *p* = 0.001, η^2^ = 0.02). Tukey’s post hoc test revealed that physical activity was significantly lower among adolescents from low-family-affluence groups than among those from high-family-affluence groups (*p* < 0,05). There was a statistically significant difference in self-rated health by family affluence (*F*(2, 807) = 8.55, *p* < 0.001, η^2^ = 0.02). Tukey’s post hoc test revealed that adolescents from medium- and high-level families rated their personal health better than those from low-level families (*p* < 0.05).

### 3.3. Factorial Validity and Reliability of HLS_19_-Q12

Before the main data analysis, the factorial validity of the HLS_19_-Q12 was evaluated. First, by conducting EFA (principal component analysis), a single-factor structure was identified (Kaiser–Meyer–Olkin = 0.85, Bartlett’s test of sphericity *χ*^2^ = 1514.66, *p* < 0.001); factor loadings ranged from 0.41–0.65 (Table 3).

Next, CFA was used to test the one-dimensional structure of the HLS_19_-Q12 determined by EFA. The results of the CFA were close to acceptable values of model fit: CFI = 0.89, TLI = 0.88, NFI = 0.89, RMSEA = 0.06. The inspection of the modification indices suggested that it induced a correlated error from item 37 to item 42. Hereby, a re-specified 12-item model increased all model fit indices: CFI = 0.93, TLI = 0.91, NFI = 0.90, RMSEA = 0.05 (0.04–0.06) (Figure 1). Figure 1 shows the resulting structural model.

The reliability of the HLS_19_-Q12 was measured by calculating Cronbach’s alpha and McDonald’s omega. It was found that the HLS_19_-Q12 has an acceptable reliability index (α = 0.885). Additionally, this scale had acceptable reliability by assessing it by McDonald’s omega (ω = 0.877).

### 3.4. General HL

General HL was calculated using the sum of the scores of all the items and standardized to a 0–100 scale. The descriptive analysis revealed that the mean score was 84.38 (SD = 18.80) for the entire sample (Table 4). Two-way ANOVA was conducted to examine the effects of gender and grade on HL scores. There was no significant interaction of gender and grade (*F* (2,801) = 1.01, *p* = 0.398, η_p_^2^ = 0.001). However, the simple main analysis showed that boys’ HL scores were higher than girls’ scores (*F* (1, 809) = 10.03, *p* = 0.002, η_p_^2^ = 0.01). Comparing HL mean scores between grades, significant differences were not found (*p* = 0.398). The categories of HL were also calculated. Analysis shows that among the entire sample, 25.5% had excellent, 41.6% sufficient, 27.7% problematic, and 5.2% inadequate levels of HL (Table 4). When comparing HL levels by gender, a statistically significant difference was found (χ^2^ (3, N = 809) = 7.83, *p* = 0.05), with a greater proportion of boys with higher levels of HL. No differences were found in the levels of HL by grade.

### 3.5. Factors Correlated with HL

In this study, we analyzed the correlation between the HLS-Q12 total score and other study variables. The results indicated that HL was positively correlated with physical activity (r = 0.14, *p* < 0.01), self-rated health (r = 0.29, *p* < 0.01), and family affluence (r = 0.08, *p* < 0.05). In addition, HL was negatively correlated with smoking during one’s lifetime (r = −0.10, *p* < 0.01) and alcohol use during one’s lifetime (r = −0.14, *p* < 0.01) as well as during the last 30 days (r = −0.09, *p* < 0.01).

### 3.6. Effects of HL on Health Behavior Indicators

The associations between HL and health behavior indicators are presented in Table 5. Multiple linear regression analysis revealed that when gender, grade, and family affluence were controlled for, HL was significantly positively associated with physical activity and explained approximately 11% of the variance (*F* (4, 802) = 25.72, *p* < 0.001). HL is also positively and significantly associated with self-rated health (*F* (4, 802) = 28.53, *p* < 0.001). Regression was also run to analyze the prediction of smoking during one’s lifetime. This resulted in a significant model (*F* (4, 802) = 14.55, *p* < 0.001) and indicated that HL was a significant negative predictor. The model predicting smoking in the past 30 days was significant (F (4, 802) = 10.95, *p* < 0.001), but HL was not a significant predictor. Both regression models for alcohol use per lifetime (*F* (4, 802) = 23.70, *p* < 0.001) and for the past 30 days (F (4, 802) = 8.56, *p* < 0.001) were significant and indicated that HL was a significant negative predictor. In all regression models, there was no evidence of multicollinearity (means of VIF = 1.01).

## 4. Discussion

This study focused on adolescents’ HL using the HLS_19_-Q12 instruments and health behavior, examining the relationship between these two items. To our knowledge, this is the first study conducted in Lithuania to examine HL and its associated factors among adolescents aged 15–19 years. Additionally, it represents the inaugural effort to assess HL in adolescents within this age group using the HLS_19_-Q12 instrument.

The HLS_19_-Q12 is a relatively recent survey instrument designed to assess HL. It aligns with the theoretical framework of HL proposed by Sorensen et al. [1] while being significantly more concise. Research indicates that adolescents often struggle to complete longer versions of the questionnaire, particularly when they are less familiar with or lack experience with the topics addressed in the statements [44]. Consequently, the shorter version may facilitate data collection, especially when additional variables need to be measured within the same survey. As we embark on research involving adolescents utilizing this survey instrument, our initial focus is to evaluate the factorial validity and reliability of this questionnaire.

Our findings showed that the HLS_19_-Q12 has good reliability and adequate structural validity. We used the CFA model to test structural validity, which aligns well with the data, suggesting that the HLS_19_-Q12 items can be effectively modeled as obvious variables of a single latent variable [46]. Our CFA results indicated that a single factor adequately represented the data, confirming that the HLS_19_-Q12 is a unidimensional tool for assessing general HL among 15–19-year-old adolescents.

It is relevant to mention that when testing the construct validity of this questionnaire with Lithuanian PE teachers, the model fit indices were slightly better with adolescent data [39]. This study confirmed the good reliability of the HLS_19_-Q12 by evaluating Cronbach’s and McDonald’s omega. Previous studies with the youth showed very similar reliability (ω = 0.86) of this instrument with a sample of young people aged 16–25 years [48]. Interestingly, in our study, the reliability was higher than that reported in a survey conducted by Pelikan and colleagues [46] in 17 European countries (ranging from 0.64–0.86) and slightly lower (0.93) than that reported by Liu and colleagues [59] regarding a survey in a Chinese adult population. In addition, this instrument showed a lower reliability coefficient in a study with Lithuanian PE teachers (α = 0.73 and ω = 0.75) [39]. This raises the question of whether there is a difference in reliability between the long version (HLS-EU-Q47) and the short version in studies with adolescents. In Lithuania, only a few studies have used the long version with pupils [60], but they did not provide reliability data. Notably, one study tested a shorter version of the HL questionnaire, specifically the HLS-EU-Q16, with adolescents in grades 8–12 [61]. The results showed that the internal consistency (McDonald’s omega coefficient) of the 16-item scale was 0.84, which is slightly lower than that in our study using HLS_19_-Q12. Hence, the results do not suggest that the HLS_19_-Q12 is superior in measuring adolescents’ HL than the HLS-EU-Q47. However, acceptable structural validity and reliability data indicate that this shorter instrument could be more suitable for examining adolescents’ HL because of its brevity and lower time-consuming costs.

After the structural validity and reliability of the HLS-19-Q12 were evaluated, the second aim of our study was to measure the general HL of adolescents. We found that the overall HL score in our sample was 84.38 ± 18.80, which is higher than that reported by other researchers in the adult population [46,59,62] and slightly different from that reported in the Lithuanian population of physical education teachers (85.09 ± 17.23) [39]. HL is a critical factor influencing individual health outcomes and can vary considerably across countries. Our survey revealed that a quarter (25.5%) of the adolescents had excellent HL, fewer than half (41.6%) had sufficient HL, fewer than one-third (27.7%) had problematic HL, and 5.2% had inadequate HL. A study conducted in Lithuania with pupils in grades 7–10 [28] revealed that a smaller proportion (17.4%) of younger pupils had excellent HL. A study in Turkey with pupils in grades 6–8 [63] showed similar results (17.7%), whereas a study conducted by Pakkari et al. [64] in Finland revealed that 34.0% of pupils aged 13–15 years achieved a high level of HL. Another study in Turkey with older adolescents (which is more in line with our research participants) revealed results that are closer to those we obtained, i.e., it was found that 21.6% had a high level of HL [42]. A study conducted in Poland with 17-year-old adolescents also found that 17.0% had a high level of HL [65]. In contrast to these findings, evidence suggests that adolescents in Germany may exhibit lower levels of HL: a study conducted by Berens et al. [66] indicated that only 10.3% of young people aged 15–29 years possess excellent HL. These differences may be attributed to variations in age, as older pupils participated in our study, as well as to different HL measurement instruments and disparities in educational systems. Specifically, in the Finnish education system, health issues are taught as a distinct subject, which is a mandatory component of both primary and secondary education, whereas in Lithuania, health education is integrated into different educational subjects. It is noteworthy that comparing our results with those of other studies that utilized the same HLS_19_-Q12, we observed that a greater proportion of respondents in our study (67.1%) demonstrated excellent and adequate HL than did the adult population across 17 European countries (55%) [46]. Additionally, the proportion of respondents in our study was slightly lower than that found in the Lithuanian population of physical education teachers, which was 68.1% [39].

In our research, boys had higher HL scores than girls. This finding aligns with earlier studies [67]. However, findings on gender differences are contradictory; some studies have indicated no gender differences [31,68], whereas others have suggested higher levels of HL for girls [25,61]. Furthermore, studies in adolescents [31,69,70] reported that there is a social gradient indicating socioeconomic differences in HL. They emphasized that the lower socioeconomic status of the family increases the likelihood that adolescents will have lower HL. Our results are very much in line with those of previous research, which shows that family affluence is significantly associated with adolescent HL [31,71,72,73].

The third aim of our research was to evaluate the associations between adolescents’ HL and health behavior determinants. Previous research has shown a link between adolescents’ HL and health behaviors [17,27,74,75,76]. In our study, we evaluated the relationships between HL and various health determinants, including physical activity, tobacco use, and alcohol consumption. Previous research has indicated a positive association between HL and physical activity [27,77,78]. The results of our study support this assertion. We identified a positive association between HL and physical activity: adolescents with higher levels of HL are more physically active.

We identified a positive relationship between HL and self-rated health. Similar findings were reported in previous studies on the effects of HL and self-rated health [26,29,30,31,72]. Our study also revealed the negative effect of HL on adolescents’ alcohol use during their lifetime and during the last 30 days. Previous studies also found that higher levels of HL are associated with lower alcohol consumption among adolescents [24,25,26]. Notably, we found a negative relationship between HL and smoking during one’s lifetime but not during the last 30 days. Some previous studies have revealed that HL is significantly negatively associated with smoking in the last 30 days [25,78]. On the other hand, these studies, unlike our study, did not assess lifetime smoking. Smoking in the past month may only reflect a temporary behavior rather than a habit or strong dependence on nicotine. Previous meta-analyses have shown that insufficient health literacy is specifically linked to nicotine dependence [79]. Other studies on youth have also indicated that the effect of health literacy on smoking habits (i.e., how long one smokes) is greater than its effect on smoking frequency [80]. On the other hand, the relationships between lifetime smoking and smoking in the last 30 days may differ across various HL domains [81]. Therefore, the data suggest that we should consider how smoking habits are assessed—whether to focus solely on recent smoking (e.g., in the last month), lifetime smoking, or both. Additionally, it is important to understand what adolescents associate with smoking.

Although we studied the associations between HL and health behavior, it is also worth discussing the data on the health behavior of adolescents. Our findings indicated that boys were more physically active and had better self-rated health than girls. At the same time, older adolescents are less physically active, are more likely to have smoked in their lifetime and during the last 30 days, and have used alcohol per lifetime and during the last 30 days. These findings align with earlier studies showing gender [23,82,83,84] and age differences [85] in physical activity and gender differences in self-rated health [31]. Our study confirms the findings of other researchers that family affluence is an important indicator of health in adolescents. Young people with low family affluence had lower levels of physical activity [86] and lower self-rated health [26]. We found no differences in alcohol consumption or smoking by gender, although other studies have reported differences when adolescents are compared by gender [87]. However, some trends show that the gap between boys and girls is narrowing [88].

In summary, this is the first study in Lithuania to explore HL using the HLS_19_-Q12 instrument and to analyze the determinants of HL and health behaviors among adolescents aged 15–19 years. The study revealed important data about adolescents’ HL. The results also confirmed that the Lithuanian version of the HLS_19_-Q12 possesses sufficient structural validity and reliability. We also examined the effects of HL on health behavior and lifestyle among adolescents.

### Limitations and Future Directions

Despite the aforementioned strengths of the study, it also has several limitations. Given that participation was voluntary, individuals who chose not to participate may have had lower HL, which raises the potential for selection bias in the study. Furthermore, although the sample size of the adolescent survey is large enough, it is not representative of the entire adolescent population in the country. It is also important to note the potential self-report biases in health behavior measurements, especially when participants are asked to report their behavior over an extended period of time. Also, since this was a cross-sectional study, the data obtained do not allow us to establish causal relationships between health literacy and health behavioral indicators.

We encourage further research to test the validity and reliability of the HLS_19_-Q12 instrument among adolescents in other countries. It would also be appropriate to assess adolescents’ digital HL when examining their HL. A recent scoping review showed increased research interest in digital HL and also found positive relations between digital HL and health-related decisions, mental health, and overall quality of life [89]. Other recent reviews have also highlighted the importance of digital HL among adolescents, particularly the integration of HL and digital HL when developing health education programs [90]. However, more research tools for measuring digital HL are available for adults compared to adolescents [91]. It is worth noting that the HLS19-Consortium of M-POHL [46,92] developed not only the HLS_19_-Q12 but also a measurement tool for digital HL (HLS_19_-DIGI). Additionally, the latter has already been validated among Norwegian adolescents [93]. Therefore, it is worth considering the use of HLS_19_-DIGI when studying both general and digital HL. This would allow us not only to measure levels of HL but also to explore how general and digital HL are related to adolescents’ healthy lifestyles. In exploring the links between HL and healthy lifestyle indicators, it would be useful to look at some lifestyle indicators in detail. For example, in the case of smoking, it would be relevant to look not only at tobacco smoking but also at vaping, which has been addressed in recent studies [94]. At the same time, it would be useful to assess how this relates to HL.

## 5. Conclusions

This study revealed that the Lithuanian version of the HLS_19_-Q12 had good reliability and adequate structural validity and can be used to assess the HL of upper-grade adolescents. Our findings indicate that only a quarter of adolescents have an excellent level of HL and that more than one-third have inadequate or poor HL. Thus, a significant proportion of adolescents need to improve their HL because it is an important determinant of health behaviors. Adolescents’ higher HL is positively associated with self-rated health and physical activity and negatively associated with smoking during one’s lifetime and alcohol use during one’s lifetime and during the last 30 days. This highlights the necessity of integrating HL into general education curricula in schools and assessing it regularly.

## Figures and Tables

**Figure 1 healthcare-12-02585-f001:**
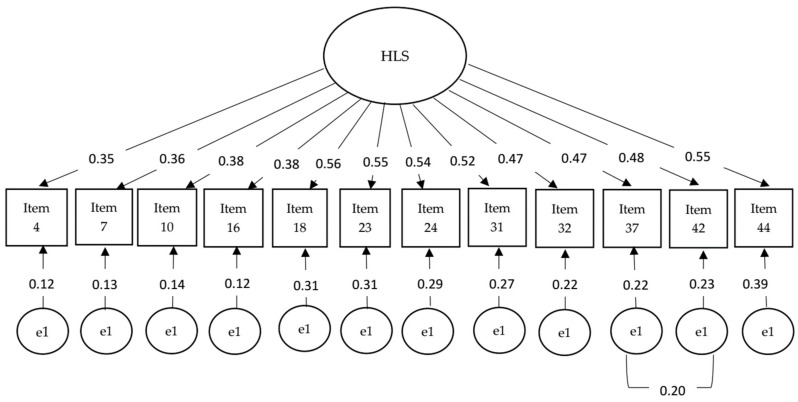
HLS_19_-Q12 model from confirmatory factor analysis.

**Table 1 healthcare-12-02585-t001:** Sample characteristics (*n* = 809).

Variable	Categories	*n*	%/M (SD)
Age		809	16.40 (1.11)
Sex	Girls	468	57.8
	Boys	341	42.2
Grades	9th grade	229	28.3
	10th grade	222	27.4
	11th grade	207	25.6
	12th grade	151	18.7
Family affluence	Low	182	22.5
	Medium	372	46.0
	High	255	31.5

Note: *n*: number of participants; %: percentage; M: mean; SD: standard deviation.

**Table 2 healthcare-12-02585-t002:** Health behavior indicators and self-rated health.

Grade/ Gender	Physical Activity	Smoking (LIFETIME)	Smoking (Past 30 Days)	Alcohol Use (Lifetime)	Alcohol Use (Past 30 Days)	Self-Rated Health
	M (SD)	M (SD)	M (SD)	M (SD)	M (SD)	M (SD)
9th grade	3.03 (2.04)	1.96 (1.85) ^11,12^	1.38 (1.21) ^11,12^	2.06 (1.66) ^11,12^	1.35 (0.92) ^12^	3.74 (0.88)
Girls	2.48 (2.05) ***	2.08 (2.02)	1.48 (1.37)	2.10 (1.75)	1.38 (0.92)	3.57 (0.85) ***
Boys	3.76 (1.80)	1.79 (1.60)	1.25 (0.93)	2.00 (1.53)	1.31 (0.93)	3.98 (0.88)
10th grade	3.21 (2.03) ^12^	2.44 (2.19) ^12^	1.70 (1.68) ^12^	2.41 (1.93) ^12^	1.53 (1.17) ^12^	3.88 (0.86)
Girls	2.85 (2.05) **	2.58 (2.30)	1.68 (1.66)	2.53 (1.88)	1.44 (0.88)	3.78 (0.87) *
Boys	3.72 (1.90)	2.24 (2.01)	1.73 (1.72)	2.24 (2.00)	1.66 (1.48)	4.02 (0.84)
11th grade	2.77 (2.05)	2.74 (2.34) ^12^	1.68 (1.87) ^12^	1.93 (2.01) ^12^	1.17 (0.88) ^12^	3.86 (0.90)
Girls	2.10 (1.76) ***	2.91 (2.49)	2.12 (2.02)	3.04 (2.16)	1.63 (0.87)	3.69 (0.88) **
Boys	3.56 (2.09)	2.54 (2.16)	1.75 (1.68)	2.59 (1.80)	1.55 (0.91)	4.05 (0.88)
12th grade	2.66 (2.07)	3.60 (2.65)	2.50 (2.15)	3.81 (2.28)	1.94 (1.07)	3.81 (0.83)
Girls	2.18 (1.95) ***	3.47 (2.67)	2.37 (2.13)	3.83 (2.22)	1.98 (1.06)	3.68 (0.85) *
Boys	3.41 (2.05)	3.81 (2.64)	2.70 (2.19)	3.79 (2.39)	1.87 (1.09)	4.02 (0.77)
All	2.94 (2.05)	2.59 (2.30)	1.82 (1.75)	2.68 (2.04)	1.57 (1.03)	3.82 (0.87)
Girls	2.43 (1.98) ***	2.69 (2.39)	1.87 (1.81)	2.79 (2.07)	1.58 (0.95)	3.68 (0.86) ***
Boys	3.64 (1.95)	2.46 (2.17)	1.76 (1.68)	2.52 (1.98)	1.56 (1.14)	4.02 (0.85)
Family affluence						
Low	2.52 (2.07) c	2.74 (2.37)	2.03 (1.90)	2.79 (2.02)	1.65 (1.13)	3.60 (0.86) b,c
Medium	2.93 (1.96)	2.55 (2.29)	1.80 (1.79)	2.62 (1.97)	1.50 (0.87)	3.85 (0.85)
High	3.25 (2.09)	2.55 (2.26)	1.69 (1.57)	2.67 (2.15)	1.62 (1.15)	3.94 (0.89)

Note: M: mean; SD: standard deviation. ^11, 12^—significant difference with the grade indicated by the respective number. b—significant difference with the medium, c—significant difference with the high family affluence group. * *p* < 0.05, ** *p* < 0.01, *** *p* < 0.001.

**Table 3 healthcare-12-02585-t003:** Factor loadings.

Items	Factor Loadings
COREHL4	0.34
COREHL7	0.44
COREHL10	0.41
COREHL16	0.46
COREHL18	0.58
COREHL23	0.60
COREHL24	0.62
COREHL31	0.57
COREHL32	0.63
COREHL 37	0.62
COREHL42	0.59
COREHL44	0.60

**Table 4 healthcare-12-02585-t004:** General HL comparison by age and gender.

	Mean	SD	Levels of HL %			
			Inadequate	Problematic	Sufficient	Excellent
9th grade	83.88	17.35	5.0	29.3	42.3	23.4
Girls	80.51	19.57	7.9	32.4	39.6	20.1
Boys	88.53	12.32	1.0	25.0	46.0	28.0
10th grade	83.77	20.75	5.4	27.8	43.0	23.8
Girls	82.72	19.94	6.1	29.5	43.2	21.2
Boys	85.3	21.88	4.4	25.3	42.9	27.5
11th grade	85.8	18.13	3.9	27.5	40.6	28.0
Girls	84.7	18.45	4.4	30.1	34.5	31.0
Boys	87.5	17.69	3.2	24.5	47.9	24.5
12th grade	83.95	19.11	7.3	26.0	39.3	27.3
Girls	82.96	18.79	7.4	26.6	40.4	25.5
Boys	85.61	19.69	7.1	25.0	37.5	30.4
Total	84.38	18.80	5.2	27.7	41.6	25.5
Girls	82.51	19.26	6.5	29.8	39.8	23.9
Boys	86.9	17.92	3.5	24.8	44.0	27.7

**Table 5 healthcare-12-02585-t005:** Multivariable linear regression models of health behavior indicators by HL and sociodemographics.

	Physical Activity	Smoking (Lifetime)	Smoking (Past 30 Days)	Alcohol Use (Lifetime)	Alcohol Use (Past 30 Days)	Self-Rated Health
Gender	0.275 ***	−0.036	−0.019	−0.043	0.008	0.158 ***
Grade	−0.070 *	0.239 ***	0.213 ***	0.292 ***	0.186 ***	0.032
Family affluence	0.110 ***	−0.004	−0.050	0.014	0.017	0.114 ***
HL	0.102 **	−0.096 **	−0.054	−0.140 ***	−0.092 **	0.265 ***
*R* ^2^	0.114	0.068	0.053	0.107	0.042	0.124
Adjusted *R*^2^	0.109	0.063	0.048	0.102	0.037	0.120

Note: Data are presented in standardized β. Independent variables: gender (1 = female; 2 = male), grade (1 = 9th; 2 = 10th; 3 = 11th; 4 = 12th), Family affluence (1 = low; 2 = medium; 3 = high), and HL score (0–100). Dependent variables: higher values indicate more frequent physical activity, better perceived personal health, and more frequent smoking and alcohol use. * *p* < 0.05, ** *p* < 0.01, *** *p* < 0.001.

## Data Availability

The data that support the findings of this study are available from the corresponding author, [S.S.], upon reasonable request.

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
