# Peer review of "Exploring the Association Between Adolescents’ Health Literacy and Health Behavior by Using the Short Health Literacy (HLS19-Q12) Questionnaire"

_healthcare, 2024, doi:10.3390/healthcare12242585_

Round 1
Reviewer 1 Report
Comments and Suggestions for Authors
The manuscript is well-prepared, with a clear focus on the validation of HLS19-Q12 and its association with health behaviors. While the study's local focus is a strength, further efforts to situate findings in a broader international context would enhance its contribution to global HL research. Constructive feedback on clarifying specific findings and expanding on non-significant results can guide improvements.
1. Introduction and Background
- Comments:
The introduction provides a comprehensive overview of health literacy (HL), its definitions, and relevance to adolescent health. However, it can be improved by including more specific references to studies validating the HLS19-Q12 for adolescents, especially comparative studies from other regions. Including such references could strengthen the context for the study's relevance in Lithuania.
2. Research Design
- Comments:
The cross-sectional, school-based design is appropriate for the research objectives. The clustered hierarchical sampling is well-suited for representing the target population. Ethical considerations and participant recruitment are clearly described.
3. Methods
- Comments:
The methods section is detailed and includes appropriate statistical analyses (e.g., EFA, CFA, regression analyses). However, a slight improvement could involve further discussion of how the pilot study informed the main study design, especially the adaptation of HLS19-Q12 for adolescents.
4. Results
- Comments:
Results are presented clearly with adequate use of tables and statistical metrics. Specific findings on gender differences in HL and its associations with health behaviors are well-detailed and contextualized. Additional discussion on non-significant findings (e.g., smoking in the past 30 days) could enhance clarity.
5. Conclusions
- Comments:
The conclusions are supported by the results. Emphasis on integrating HL into education curricula is valid. Adding actionable recommendations for further research, such as testing digital HL, would improve this section.
Reviewer 2 Report
Comments and Suggestions for Authors
Comprehensive Manuscript Evaluation Report
General Considerations:
The manuscript presents a significant and well-conducted study on health literacy (HL) among Lithuanian adolescents, utilizing the HLS19-Q12 instrument. This research makes a substantial contribution to the field, offering valuable insights into the relationship between HL and health behaviors in adolescents.
General Strengths:
- Pertinent and timely topic in the realm of public health and health education.
- Robust methodology, including instrument validation and appropriate statistical analyses.
- Substantial participant sample.
- Comprehensive approach, considering multiple health behaviors.
General Weaknesses:
- Study limitations could be more thoroughly delineated and emphasized.
- Certain thematic areas in the references could be more extensively explored.
General Improvement Suggestions:
- Expand the discussion on the study's limitations.
- Further update and diversify the bibliographic references.
Manuscript Section Evaluations:
- Objectives:
Assessment: ADEQUATE
Strengths:
- Clearly defined objectives, coherent with the title and investigated problem.
- Encompass instrument validation, HL measurement, and analysis of relationships with health behaviors.
Weaknesses: No significant weaknesses identified.
Improvement Suggestions: None necessary.
- Methodology:
Assessment: ADEQUATE
Strengths:
- Methodological strategy appropriate to the objectives.
- Clear description of participants, data collection procedures, and data analysis.
- Detailed approach to the psychometric qualities of the HLS19-Q12.
- Adequate mention of original sources for instruments.
Weaknesses: No significant weaknesses identified.
Improvement Suggestions: None necessary.
- Data Analysis:
Assessment: ADEQUATE
Strengths:
- Analysis well-articulated with objectives and theoretical framework.
- Appropriate and well-executed statistical techniques.
- Clear and organized presentation of results.
Weaknesses: No significant weaknesses identified.
Improvement Suggestions: None necessary.
- Ethical Aspects:
Assessment: ADEQUATE
Strengths:
- Clear mention of Ethics Committee approval.
- Adequate consideration of ethical aspects, including informed consent and confidentiality.
- Compliance with international ethical guidelines.
Weaknesses: No significant weaknesses identified.
Improvement Suggestions: None necessary.
- Results:
Assessment: ADEQUATE
Strengths:
- Clear and objective presentation of main results.
- Effective use of tables and figures.
- Appropriate emphasis on important findings.
Weaknesses: No significant weaknesses identified.
Improvement Suggestions: None necessary.
- Discussion:
Assessment: ADEQUATE
Strengths:
- Pertinent and sufficient discussion of results.
- Good contextualization within existing literature.
- Consideration of practical implications and study limitations.
Weaknesses: No significant weaknesses identified.
Improvement Suggestions: None necessary.
- Comparison between Results and Literature:
Assessment: ADEQUATE
Strengths:
- Effective integration of results with existing literature.
- Appropriate use of theoretical concepts.
- Nuanced discussion of similarities and differences with previous studies.
Weaknesses: No significant weaknesses identified.
Improvement Suggestions: None necessary.
- Conclusions:
Assessment: ADEQUATE
Strengths:
- Clear conclusions well-supported by results.
- Effective synthesis of main findings.
- Consideration of practical implications and future directions.
Weaknesses: No significant weaknesses identified.
Improvement Suggestions: None necessary.
- Contributions and Limitations:
Assessment: ADEQUATE WITH RESERVATIONS
Strengths:
- Study contributions well-described throughout discussion and conclusions.
- Mention of some important limitations in the discussion section.
Weaknesses:
- Limitations not explicitly highlighted at the end of the manuscript.
- Some potential limitations not addressed.
- Lack of in-depth discussion on the implications of limitations.
Improvement Suggestions:
- Add a specific "Limitations and Future Directions" subsection at the end of the discussion or before the conclusion.
- Expand the discussion of limitations to include: a) Cross-sectional nature of the study and limitations for causal inferences. b) Potential self-report biases in health behavior measures. c) Specific limitations of the HLS19-Q12 instrument.
- Explicitly discuss how identified limitations could be addressed in future research.
- Balance the discussion of limitations with a concise summary of the study's main strengths and contributions.
- References:
Assessment: ADEQUATE WITH RESERVATIONS
Strengths:
- Pertinent and comprehensive references.
- Good mix of theoretical studies, empirical research, and systematic reviews.
- Inclusion of international and Lithuania-specific sources.
Weaknesses:
- Proportion of recent references (last 5 years) close to, but not reaching, the ideal 50%.
- Some thematic areas could be more extensively explored.
- Possible lack of geographical balance in references.
Improvement Suggestions:
- Include more studies published in the last two years (2022-2024) to increase the proportion of recent references.
- Add more references from research conducted in Eastern European countries or those with socioeconomic contexts similar to Lithuania.
- Incorporate more studies specifically addressing digital HL in adolescents.
- Include recent meta-analyses or systematic reviews on HL in adolescents and its relationship with health behaviors.
This detailed report provides a comprehensive overview of the manuscript evaluation, highlighting its significant strengths and specific areas for potential improvement, particularly in the Contributions and Limitations and References sections. The manuscript demonstrates considerable merit in its approach to investigating health literacy among Lithuanian adolescents, with robust methodology and clear presentation of results. The suggested enhancements, particularly in explicating study limitations and diversifying references, would further elevate the scholarly impact of this valuable contribution to the field of adolescent health literacy research.
